# Vacuum Ultraviolet Absorption Spectroscopy Analysis of Breath Acetone Using a Hollow Optical Fiber Gas Cell

**DOI:** 10.3390/s21020478

**Published:** 2021-01-12

**Authors:** Yudai Kudo, Saiko Kino, Yuji Matsuura

**Affiliations:** Graduate School of Biomedical Engineering, Tohoku University, 6-6-05 Aoba, Sendai 980-8579, Japan; yudai.kudo.s4@dc.tohoku.ac.jp (Y.K.); kino@ecei.tohoku.ac.jp (S.K.)

**Keywords:** breath acetone measurement, vacuum ultraviolet spectroscopy, hollow optical fiber

## Abstract

Human breath is a biomarker of body fat metabolism and can be used to diagnose various diseases, such as diabetes. As such, in this paper, a vacuum ultraviolet (VUV) spectroscopy system is proposed to measure the acetone in exhaled human breath. A strong absorption acetone peak at 195 nm is detected using a simple system consisting of a deuterium lamp source, a hollow-core fiber gas cell, and a fiber-coupled compact spectrometer corresponding to the VUV region. The hollow-core fiber functions both as a long-path and an extremely small-volume gas cell; it enables us to sensitively measure the trace components of exhaled breath. For breath analysis, we apply multiple regression analysis using the absorption spectra of oxygen, water, and acetone standard gas as explanatory variables to quantitate the concentration of acetone in breath. Based on human breath, we apply the standard addition method to obtain the measurement accuracy. The results suggest that the standard deviation is 0.074 ppm for healthy human breath with an acetone concentration of around 0.8 ppm and a precision of 0.026 ppm. We also monitor body fat burn based on breath acetone and confirm that breath acetone increases after exercise because it is a volatile byproduct of lipolysis.

## 1. Introduction

The exhaled breath of a living body contains hundreds of minute volatile organic compounds (VOCs), the components and concentrations of which reflect the body’s metabolism [1,2,3]. Therefore, by analyzing exhaled breath, useful information can be obtained for the diagnosis of various diseases and the overall management of general health. However, since VOC concentration is extremely small (usually less than 1 ppm), a high-sensitivity gas chromatography–mass spectrometry (GC-MS) system is widely used for analysis [4,5,6]. Unfortunately, these devices are usually large and expensive; thus, they are impractical for hospital and clinic use. Some groups have proposed to use micro electro-mechanical systems (MEMS)-based GC systems [7,8]. These miniaturized systems provide short analysis time and low power consumption; however, they still need the difficult preprocessing of breath samples.

A variety of optical analysis methods have been proposed to solve the above problems [9,10,11]. Optical methods are usually advantageous over GC-MS methods with respect to size and cost. Moreover, optical methods usually enable analysis in real time, because the sample preparation processes required in GC-MS methods [12,13] are not needed for optical methods. Breath analysis methods utilizing Raman spectroscopy [14,15], photoacoustic spectroscopy with near-infrared and mid-infrared (MIR) light [16,17], and MIR absorption spectroscopy [18,19] have also been proposed. Some breath analysis methods using ultraviolet (UV) spectroscopy have also been proposed. One of the advantages of UV spectroscopy over MIR spectroscopy is that it is less affected by water vapor, and, therefore, a desiccant is not necessary. Additionally, most volatile gases show relatively large absorption in the UV region. Our group previously proposed a breath analysis method based on UV absorption spectroscopy using hollow optical fiber as a microvolume/long-optical path gas cell [20,21]. The proposed measurement system, composed of a laser-driven light source emitting a broadband spectrum in the UV region, a hollow optical fiber gas cell, and a fiber-coupled compact spectrometer, is both simple and cost effective. By using this system, we successfully quantitated isoprene in breath, the minimum detectable concentration of which was <100 ppb [21].

One of the VOCs in breath that can be detected by UV absorption spectroscopy is acetone. Acetone is a substance produced in the blood by the decomposition and combustion of body fat. The concentration of acetone in exhaled breath has a high correlation with the concentration of acetone in the blood, and it is known as a biomarker for diabetes [22,23,24,25]. Wang et al. proposed a cavity ring-down spectroscopy system for analyzing acetone in breath using a Q-switch Nd:YAG laser operating at 266 nm [26,27,28,29]. The estimated minimum detectable concentration of acetone gas was reportedly 57 ppb [29]. However, the system is complicated and expensive. Li et al. proposed a compact analysis system for breath acetone using a cheap light-emitting diode that can produce 285 nm of UV light [30]. They also developed a compact multipath gas cell to enhance the sensitivity. However, the lowest detectable concentration was limited to 0.7 ppm. One of the reasons for the limitation of the sensitivity is the small absorption coefficient of acetone in the UV region. For this reason, in our previous system [21], the detectable VOC was limited to only isoprene, although an advantage of gas analysis methods based on absorption spectroscopy is the simultaneous detection capability of multiple gases.

In this paper, we primarily focus on the absorption peak of acetone in the vacuum UV (VUV) region at a wavelength of approximately 195 nm. Because the absorption of acetone in the VUV region is much stronger than that in the UV region, i.e., approximately 265 nm, highly sensitive detection of breath acetone is expected. To detect the absorption peak in the VUV region, we built a measurement setup that comprises a deuterium lamp source, a hollow-core fiber gas cell, and a fiber-coupled compact spectrometer corresponding to the VUV region. The target wavelength of 195 nm is essentially the longer edge of the VUV region, and the air absorption has little effect; therefore, the vacuum components that make the system large and expensive are not necessary. In addition to the simultaneous detection capability of multiple gases that is an advantage of the spectroscopic system over conventional semiconductor sensors, the optical measurement does not need periodic calibration, which is usually necessary for conventional systems. As far as we know, this is the first successful detection of the absorption peak of breath acetone in the VUV, and this enables the simultaneous detection of acetone and isoprene in breath.

For the measured absorption spectra of human breath, we applied multiple regression analysis to quantitate the breath acetone using the absorption spectra of oxygen, water, and acetone as explanatory variables. This research aims to develop an optical spectroscopy system for the measurement of breath acetone. We report the experimental results to evaluate the sensitivity and accuracy of the system using the standard addition method based on the human breath of healthy adults. The results of monitoring the body fat burn are also discussed.

## 2. Experimental Setup and Method

Figure 1 shows a schematic of our experimental setup. As a light source, a 33 W deuterium lamp (L9519, Hamamatsu Photonics, Hamamatsu, Japan) with a synthetic silica window emitting a wavelength of 160–400 nm was used. Emitted light was directly coupled to a hollow optical fiber gas cell, the input end of which was set at a distance of 30 mm from the arc point. The input end of the fiber gas cell was capped by a metal sleeve with a calcium fluoride (CaF_2_) window and a gas inlet. The output end of the fiber gas cell was kept open, and the output light was focused by a CaF_2_ lens with a focal length of 25 mm onto a short hollow optical fiber tip (inner diameter: 1 mm; length: 100 mm) connected to a fiber-coupled spectrometer (Maya2000 Pro, Ocean Insight, Orlando, FL, USA). The spectrometer was equipped with a back-thinned charged-coupled device image sensor, the wavelength range and resolution of which was 80–300 nm and 0.22 nm, respectively. In the experiment, the spectrometer and the coupled short fiber tip were purged with nitrogen.

We used an aluminum-coated hollow optical fiber (UVS1000, Doko Engineering LLC, Sendai, Japan) as a gas cell. The base material of the hollow optical fiber was a silica-glass tube, and the inner surface of the tube was coated with an aluminum film using metal organic chemical vapor deposition [31]. Figure 2 shows the loss spectra of the 1-m-long aluminum-coated hollow-glass optical fiber with two different inner diameters. The hollow core of the fibers was purged with nitrogen. One can see that losses at wavelengths longer than 190 nm are almost constant, although the losses abruptly increase in the VUV region. We adopted a fiber with an inner diameter of 2 mm because the coupling efficiency of the light emitted from the incoherent lamp source was much higher in the larger hollow core. In the experiment, we connected three hollow optical fibers (1-m long) to ensure high sensitivity. Although the fibers were kept straight in our experiment, they can be folded or looped for the future practical system. The factor that determined the long-term stability of the measurement system is the hollow optical fibers that are coated with thin aluminum film. However, we confirmed that the optical transmission properties of the fibers did not change after being repeatedly used for the breath measurement for more than one year.

For the measurement of human breath, the breath-collection bags schematically shown in Figure 3 were used. Although Tedler^®^ bags are often used for gas sampling owing to the air tightness and chemical stability [32], we used simple polyethylene Ziploc^®^ bags because they are easy to obtain at low cost. As the optical measurement was performed immediately after the breath sampling in our experiment, we found no problem in the leak or degradation of the breath samples. At the beginning of exhalation, breath was collected in a large plastic bag (A), because this part of the breath is not involved in the gas exchange process in the alveoli. Then, the valve was switched, and the terminal exhaled breath that received blood gas in the alveoli was collected in a small bag (B), which was used as a sample. Our protocol was approved by the ethical committee on the Use of Humans as Experimental Subjects of Tohoku University (No. 20A-7), and informed consent was obtained from the examinees.

## 3. Results

### 3.1. Measurement of Standard Acetone Gases

We measured the absorption spectra of acetone gases in known concentrations. Figure 4 shows the absorption spectrum of 10 ppm acetone in the VUV and UV regions. From this result, we found that the peak height in the VUV region at approximately 195 nm is more than 500 times larger than that in the UV region. Accordingly, we can expect a highly sensitive measurement of acetone by detecting absorption in the VUV region.

Figure 5 shows the absorption spectra of acetone with different concentrations. These gases were generated by diluting the 10 ppm acetone standard gas with nitrogen; the transmission of pure nitrogen was used as a background. A clear peak can be observed at 195.1 nm, even for the concentration of 0.1 ppm. Although another two absorption peaks are seen in Figure 4 at 187.0 and 191.0 nm in the VUV region, we focused on only the peak at 195.1 nm because the other peaks in short wavelengths were more strongly affected by the absorption of oxygen.

### 3.2. Measurement of Human Breath

We collected breath samples from five healthy adults aged between 23 and 24 years. In the experiment, we did not restrict exercise or diet. Figure 6 shows an absorption spectrum of a sample measured by the proposed system compared with those of oxygen and acetone. We investigated the absorption spectra of known VOCs in human breath and found that no component other than acetone showing detectable absorption in this wavelength region was present. We confirmed this because the absorption spectra of exhaled breath represented by the one in Figure 6 show no effect of components other than oxygen, water, and acetone. Although the shape of the breath absorption spectrum is similar to that of oxygen, slight differences can be observed near the absorption peak of acetone. The concentration of acetone in the figure is 0.9 ppm, and we cannot quantitate the oxygen concentration because it is partly converted to ozone by irradiation with strong UV light.

Figure 7 shows a measured breath spectrum and a spectrum obtained by multiple regression analysis using oxygen and water as explanatory variables. Although nitrogen and carbon dioxide are also the main components of exhaled breath, we found that these gases show little absorption in the VUV region; therefore, we did not use these components in our analysis. We utilized the Analysis ToolPak in Excell, and 57 data points between 194–200 nm were analyzed. The differences can be observed at approximately 194.8 and 196.4 nm; they decrease by adding acetone as another explanatory variable, as shown in Figure 8. In the result with water and oxygen shown in Figure 7, the correlation coefficient R^2^ is 0.9816, which increases to 0.9926 by considering acetone as another component of breath. As a result of multiple regression analysis shown in Figure 8, the concentration of acetone in breath was estimated to be 0.46 ppm, which is consistent with the general concentration range of healthy adults (0.3–1.2 ppm) [33].

Figure 9 shows the estimated breath acetone concentrations for the five subjects. Five consecutive measurements were performed for each subject, and the dots and error bars in the figure show the averages and variations of the measured values, respectively. We confirmed that the estimated values are in the range of healthy adults, that is, 0.3–1.2 ppm [33]. The measurement precision had a standard deviation of 0.026 ppm, and, because of this small error, we could identify the individual differences.

Next, we evaluated the sensitivity and accuracy of the system using the standard addition method based on human breath. We prepared breath samples with different acetone concentrations by diluting the 10 ppm standard gas with breath samples. Figure 10 shows the absorption spectra of exhaled and simulated breath with 1.84 ppm acetone added to the original breath sample. One can see that, in the spectrum of the simulated breath sample, the shape of the absorption peak at roughly 195 nm is more affected by the absorption peak of acetone. For this simulated breath sample, the estimated acetone concentration was 2.64 ppm. Since the acetone concentration of the original breath sample was 0.82 ppm, the estimated value was determined to be approximal to the reference value of 2.66 ppm.

### 3.3. Body Fat Burn Monitoring

Recently, it has been reported that acetone in human breath correlates with the rate of fat loss in healthy people. Since acetone is a volatile byproduct of lipolysis, a strong correlation reportedly exists between the breath acetone concentration and the rate of fat loss [34]. To confirm the feasibility of our proposed system for fat burn monitoring, we tested a healthy adult volunteer during exercise and rest. The subject was asked in advance to fast for 12 h before the test to promote fat burning. In the experiment, the subject walked upstairs and downstairs for 30 min and then rested for 15 min. This exercise–rest set was repeated three times. Breath measurements were performed before the first exercise set, during the rests, and after the last exercise set. Figure 11 shows the measured acetone concentration for the subject who took the exercise test. The figure also shows breath acetone concentration changes of the same subject during rest with no exercise for comparison purposes.

## 4. Discussion

### 4.1. Measurement of Standard Acetone Gases

Figure 12 shows a correlation plot between the measured optical absorption at 195.1 nm and the concentration of acetone, from which we can confirm a linear correlation between them in the concentration range of 0.1–10 ppm; the correlation coefficient R^2^ is high as 0.9986. We calculated the lowest detection limit from the signal-to-noise ratio (SNR) of our measurement system. The noise level seen in Figure 5b is 0.007 dB. From the correlation plot in Figure 12, the concentration that corresponds to 0.021 dB that is defined by SNR = 3 is 0.024 ppm, and this is the lowest detection limit of our measurement system. Table 1 summarizes the performance of various systems in literature and the system proposed in this paper.

### 4.2. Measurement of Human Breath

Figure 13 shows the correlation between the acetone concentration estimated by the optical absorption and that of the breath samples with additional acetone. For the simulated samples, 0.46, 0.94, and 1.84 ppm of acetone were added to the original breath samples. The estimated results for the original breath samples measured before adding acetone are also plotted in the figure. The result shows high linearity, and the correlation coefficient R^2^ is 0.9995. We found that the measurement accuracy has a standard deviation of 0.074 ppm, which is a better value compared with the UV absorption measurement reported by Smeeton [30] (0.2 ppm). For healthy adults, the concentration of breath acetone is between 0.3 and 1.2 ppm [33], whereas, for diabetics, it is 1–10 ppm [36]. Accordingly, the sensitivity and accuracy of the proposed system are sufficient for the measurement of acetone in human breath.

### 4.3. Body Fat Burn Monitoring

As shown in Figure 11, the breath acetone gradually increased after the subject exercised; without exercise, the increase did not occur. This trend is consistent with the result reported by Güntner [37], although the amount of increase was much higher in our case. This may be due to individual differences because the subject in our experiment was relatively lean. From the results shown in Figure 11, it is evident that there was almost no change in the acetone concentration during exercise. This result is also consistent with the results obtained by Güntner [37]. Although it was not our purpose to investigate the mechanism of body fat burn, our proposed system is sufficient for the said application. Additionally, our system provides measurements in real time, and it can monitor the VOC concertation changes in breath, as shown in our previous paper [21]. In particular, the VUV spectroscopy system simultaneously analyzes acetone and isoprene in exhaled breath, and it can be used to further investigate the correlation between the said VOCs and the human metabolism process.

## 5. Conclusions

We proposed a VUV spectroscopy system to measure acetone in human breath using a hollow optical fiber transmitting VUV and UV light. Highly sensitive detection of breath acetone was enabled by detecting the absorption peak of acetone in the VUV region at the wavelength of roughly 195 nm, which is much larger than the peaks in the UV region of approximately 265 nm. In our spectroscopy system, we did not use the vacuum components responsible for making conventional systems large and expensive. This is because the target wavelength of 195 nm is almost the longer edge of the VUV region, and the air absorption has little effect.

We developed a measurement setup consisting of a deuterium lamp source, a hollow-core fiber gas cell, and a fiber-coupled compact spectrometer corresponding to the VUV region to measure the absorption spectra of human breath. We first measured the absorption spectra of water, oxygen, and acetone standard gas with known concentrations. Then, we applied multiple regression analysis using the said spectra as explanatory variables to quantitate the concentration of acetone in breath. To evaluate the sensitivity and accuracy of the system, we applied the standard addition method based on human breath. The evaluation shows that the measurement accuracy had a standard deviation of 0.074 ppm for healthy human breath with an acetone concentration of roughly 0.8 ppm. We also monitored body fat burn based on breath acetone. It was confirmed that breath acetone increased after exercise because it is a volatile byproduct of lipolysis.

One of the advantages of the proposed VUV spectroscopy system over conventional semiconductor sensors is that it simultaneously detects multiple gas types, including acetone and isoprene. The proposed system detects the said gases in real time and does not need any preprocessing. Moreover, the optical measurement does not need periodic calibration, which is usually necessary for conventional systems.

## Figures and Tables

**Figure 1 sensors-21-00478-f001:**
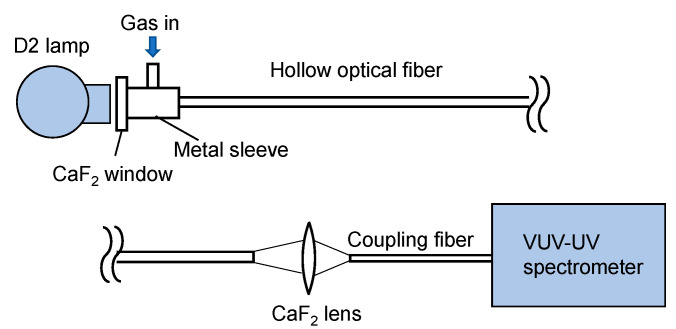
Schematic of the measurement setup.

**Figure 2 sensors-21-00478-f002:**
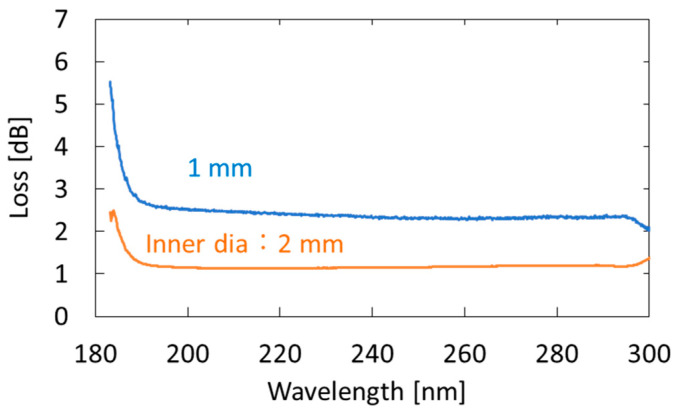
Measured loss spectra of 1-m-long aluminum-coated hollow-glass optical fibers with different inner diameters.

**Figure 3 sensors-21-00478-f003:**
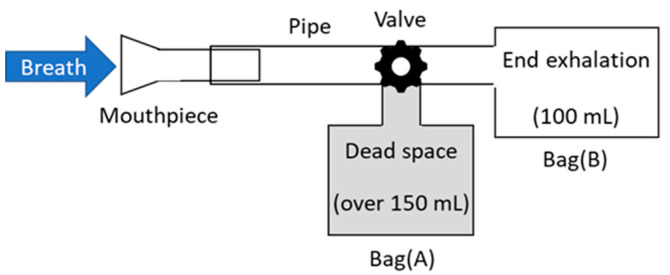
Schematic of breath-collection bags.

**Figure 4 sensors-21-00478-f004:**
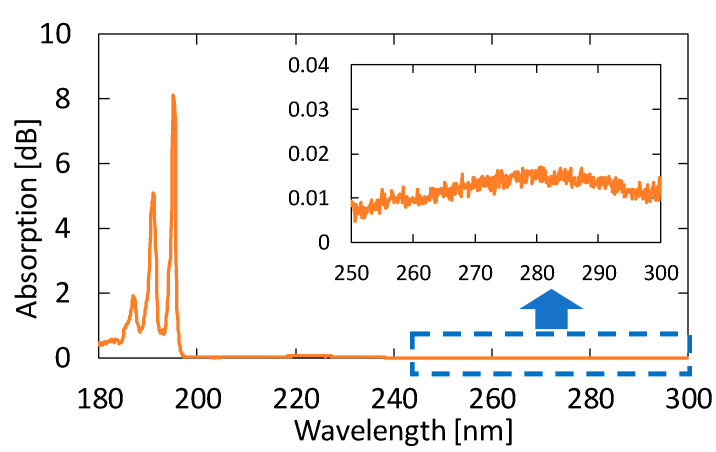
An absorption spectrum of 10 ppm acetone in the vacuum ultraviolet (VUV) and UV regions. The inset is an enlarged spectrum at approximately 280 nm.

**Figure 5 sensors-21-00478-f005:**
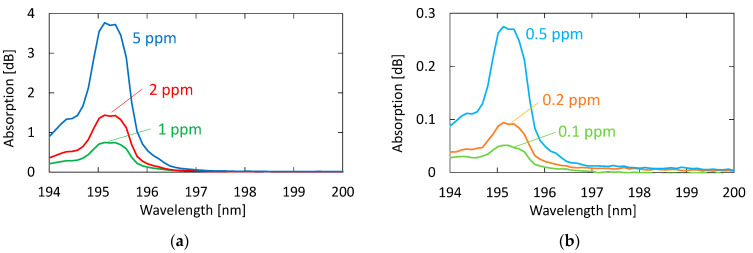
Absorption spectra of acetone with different concentrations: (**a**) Concentrations higher than 1 ppm; (**b**) concentrations lower than 0.5 ppm.

**Figure 6 sensors-21-00478-f006:**
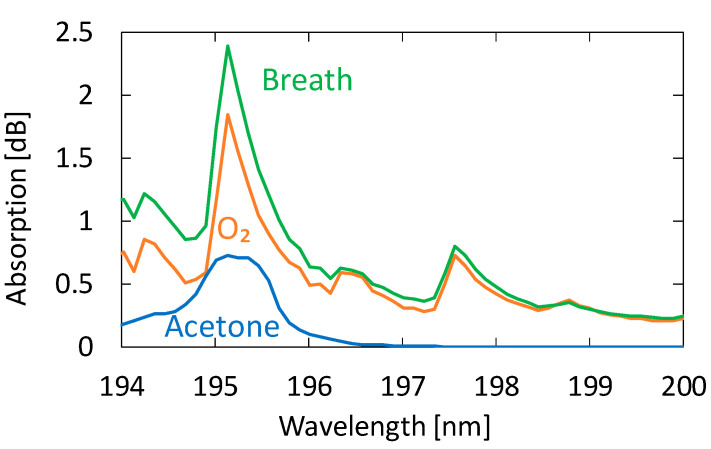
An absorption spectrum of a human breath sample measured by the proposed system compared with those of oxygen and acetone.

**Figure 7 sensors-21-00478-f007:**
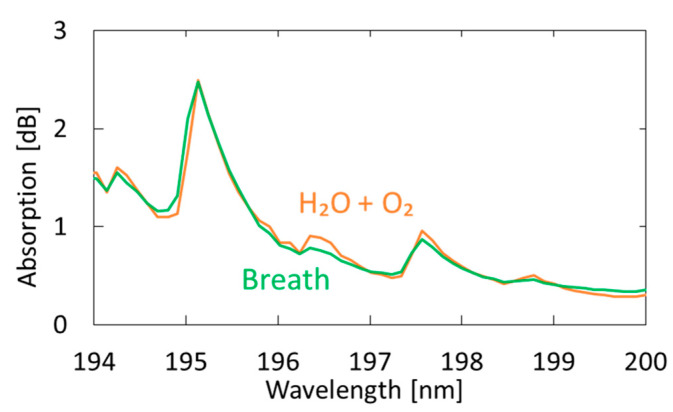
Measured spectrum of human breath compared with a spectrum obtained by multiple regression analysis using oxygen and water as the explanatory variables.

**Figure 8 sensors-21-00478-f008:**
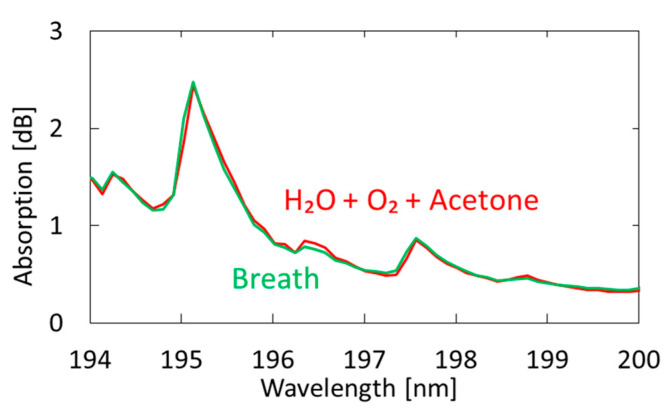
Measured spectrum of human breath compared with a spectrum obtained by multiple regression analysis using oxygen, water, and acetone as explanatory variables.

**Figure 9 sensors-21-00478-f009:**
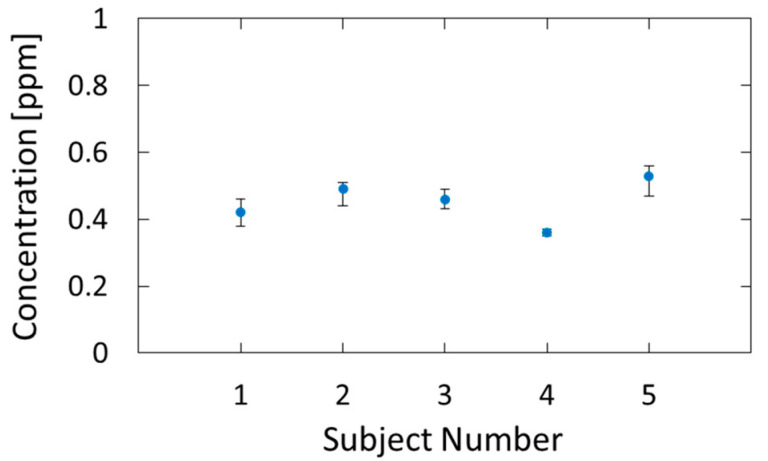
Breath acetone concentrations of five subjects estimated by the proposed method.

**Figure 10 sensors-21-00478-f010:**
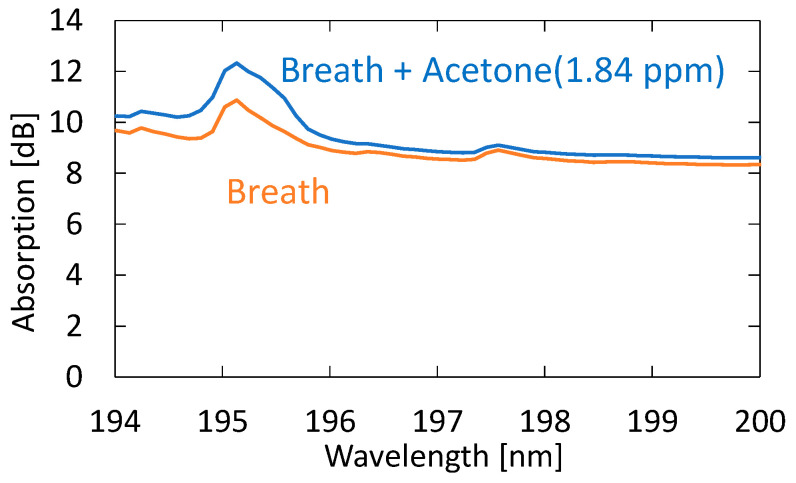
Absorption spectra of exhaled and simulated breath with 1.84 ppm acetone added to the original breath sample.

**Figure 11 sensors-21-00478-f011:**
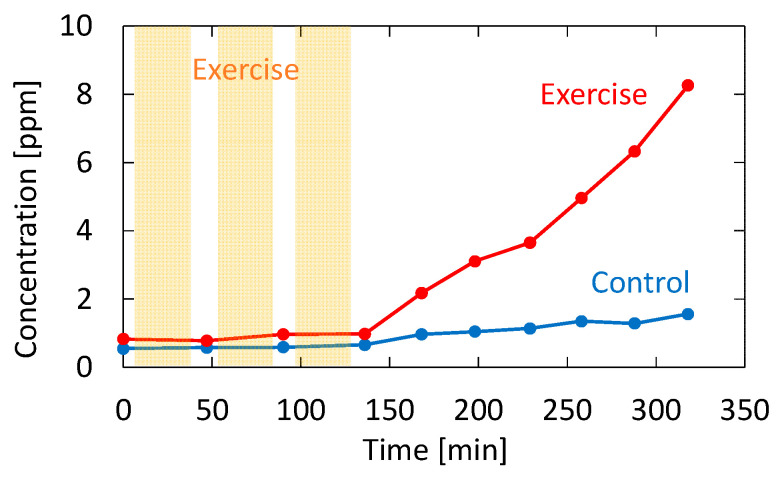
Measured acetone concentration for the subject who took the exercise test. The result for the same subject sitting still without performing exercise is also shown as a control.

**Figure 12 sensors-21-00478-f012:**
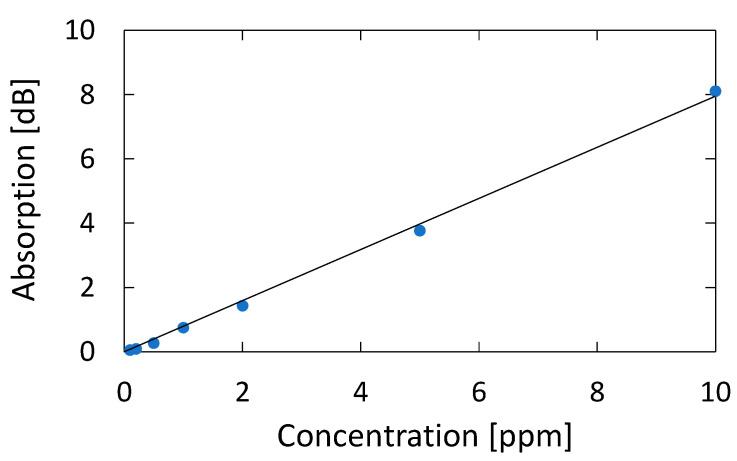
Correlation plot between the measured optical absorption at 195.1 nm and the concentration of acetone.

**Figure 13 sensors-21-00478-f013:**
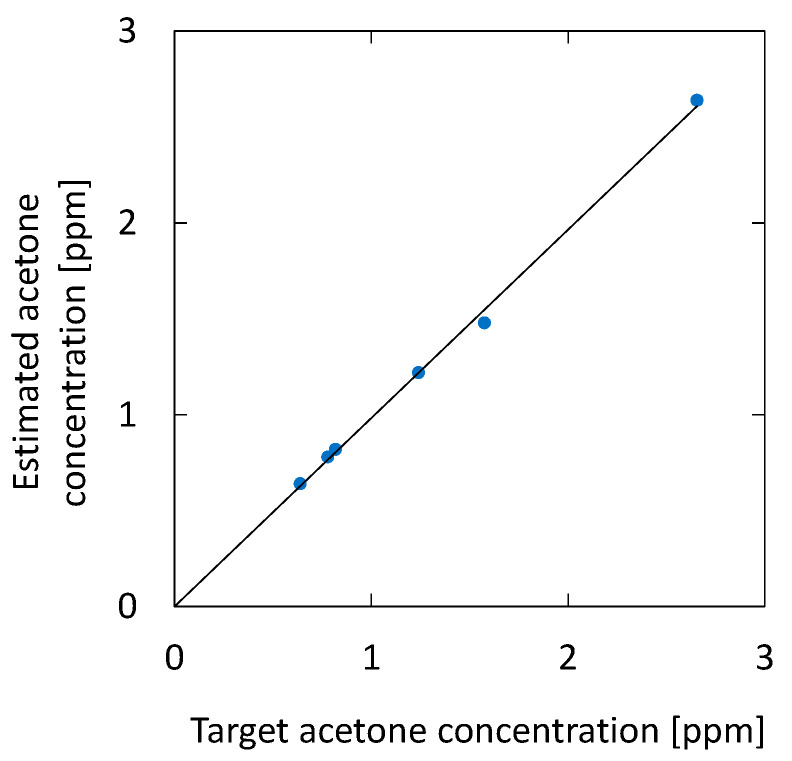
Correlation between the acetone concentration estimated by the optical absorption and that of breath samples with additional acetone. The estimated results for the original breath samples measured before adding acetone are also plotted.

**Table 1 sensors-21-00478-t001:** Comparison of the performance of various systems in literature and the system proposed in this paper.

Measurement Method	Target Components	Limit of Detection	Ref.
MEMS-based micro GC	Acetone	50 ppb	[7]
Fiber-enhanced Raman spectroscopy	CH_4_, CO_2_, N_2_O	Sub-ppm	[15]
Photoacoustic spectroscopy with UV-LED	Acetone	80 ppb	[17]
CRDS with Q-sw YAG laser (266 nm)	Acetone	57 ppb	[29]
Spectroscopy with folded-path gas cell and UV-LED (285 nm)	Acetone	0.7 ppm	[30]
SnO_2_ semiconductor sensor	Acetone	0.5 ppm	[35]
This work (VUV spectroscopy with hollow-core-fiber gas cell)	Acetone, Isoprene	24 ppb	

## Data Availability

The data presented in this study are available on request from the corresponding author. The data are not publicly available due to privacy restrictions.

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
