# Peer review of "Vacuum Ultraviolet Absorption Spectroscopy Analysis of Breath Acetone Using a Hollow Optical Fiber Gas Cell"

_sensors, 2021, doi:10.3390/s21020478_

Round 1

Reviewer 1 Report

In this work, the authors report about a vacuum ultraviolet (VUV) spectroscopy system for the measurement of acetone in human breath, based on a hollow core optical fiber for the transmission of light at those wavelengths and as a gas cell.

The following issues need to be addressed by the authors:

  • The novelty of the proposed system is not completely clear as the authors have already reported about a spectroscopy system based on hollow fiber in [19] for the detection of isoprene. Is the novelty related to the application? Which are the additional challenges involved in the current case? If none, the work should be considered as incremental and not novel enough.
  • Additional details regarding the hollow core fiber should be reported, as for example the model and manufacturer, the geometrical features of core and cladding, presence of holes, etc.
  • The authors mention regression analyses but they should provide additional details. Please clarify.
  • I suggest comparing the performance of the system with other solutions from literature by using a table.

Author Response

Thank you very much for your comments.

Please see the attached PDF file for our answers to your comments.

Reviewer 2 Report

This manuscript presents a VUV spectroscopy system for the detection of acetone in exhaled human breath. The performance was tested using healthy human breath and preliminary results were obtained. This work is interesting and could potentially contribute to the development of approaches for breath analysis. However, there are some points needed to be addressed before it is suitable for publication.

1. Since acetone is a biomarker for diabetes, it is desirable to test the breath sample collected from patients with diabetes. 

2. What is the main advantages of using this VUV spectroscopy system as compared with other approaches for breath analysis?

3. It is not clear why the oxygen, water, and acetone were utilized as explanatory variables? 

4. How about the specificity of this VUV spectroscopy system? Is there any influences of other VOCs in the breah sample on the measurement?

5. The details about the determination or calculation of sensitivity and detection limit should be presented.

6. How about the long-term stability of the system for breath analysis?

Author Response

(The authors gave the same response as above.)

Reviewer 3 Report

The manuscript presents a novel technique to measure the concentration of acetone in exhaled breath. The subject is interesting, and it is one to which the authors have made significant contributions. The experimental section is explained very well in details and the results are fully understandable for the readers. The authors have done lots of experiments and reported a significant amount of scientific data which make the paper suitable for publication.

Author Response

We appreciate very much for your comment. It helps us a lot for our future work.

Reviewer 4 Report

I found this article very well written and authors have also properly explained their results. I would advise authors to make a few minor changes to further improve quality of their manuscript.

  1. Figure 3: Typically Tedlar bags are used for human breath sampling. This is to ensure that sample does not stick to surface of bag. Did authors use Tedlar bag or a simple bag.
  2. Introduction does not talk about use of micro-GC for analysis of human breath analysis. I would recommend authors include relevant papers in literature.
  3. Figure  4: There are three separate peaks in absorption spectrum of acetone between 180nm to 200nm. Rest of the results show a single broad peak of acetone in all other results. Can authors explain this ?  

Author Response

(The authors gave the same response as above.)

Round 2

Reviewer 1 Report

The authors have addressed all the comments provided by the reviewers. The revised manuscript is suitable for publication.